# Transgenerational Inheritance of Environmentally Induced Epigenetic Alterations during Mammalian Development

**DOI:** 10.3390/cells8121559

**Published:** 2019-12-03

**Authors:** Louis Legoff, Shereen Cynthia D’Cruz, Sergei Tevosian, Michael Primig, Fatima Smagulova

**Affiliations:** 1Univ Rennes, Inserm, EHESP, Irset (Institut de recherche en santé, environnement et travail)—UMR_S 1085, F-35000 Rennes, France; louis.legoff@inserm.fr (L.L.); shereen-cynthia.d-cruz-benard@inserm.fr (S.C.D.); michael.primig@inserm.fr (M.P.); 2University of Florida, Department of Physiological Sciences Box 100144, 1333 Center Drive, Gainesville, FL 32610, USA; stevosian@ufl.edu

**Keywords:** transgenerational inheritance, epigenetics, environmental factors, genome reprograming, histone modifications

## Abstract

Genetic studies traditionally focus on DNA as the molecule that passes information on from parents to their offspring. Changes in the DNA code alter heritable information and can more or less severely affect the progeny’s phenotype. While the idea that information can be inherited between generations independently of the DNA’s nucleotide sequence is not new, the outcome of recent studies provides a mechanistic foundation for the concept. In this review, we attempt to summarize our current knowledge about the transgenerational inheritance of environmentally induced epigenetic changes. We focus primarily on studies using mice but refer to other species to illustrate salient points. Some studies support the notion that there is a somatic component within the phenomenon of epigenetic inheritance. However, here, we will mostly focus on gamete-based processes and the primary molecular mechanisms that are thought to contribute to epigenetic inheritance: DNA methylation, histone modifications, and non-coding RNAs. Most of the rodent studies published in the literature suggest that transgenerational epigenetic inheritance through gametes can be modulated by environmental factors. Modification and redistribution of chromatin proteins in gametes is one of the major routes for transmitting epigenetic information from parents to the offspring. Our recent studies provide additional specific cues for this concept and help better understand environmental exposure influences fitness and fidelity in the germline. In summary, environmental cues can induce parental alterations and affect the phenotypes of offspring through gametic epigenetic inheritance. Consequently, epigenetic factors and their heritability should be considered during disease risk assessment.

## 1. Introduction

Information not encoded in the DNA sequence is termed the epigenetic code. Epigenetics is defined as a molecular process that establishes a phenotype whereby the resulting changes persist in the absence of the original cause (e.g., [1]). At the molecular level, any changes in gene expression and the transmission of phenotypic variations to subsequent generations that do not result from alterations in the DNA sequence are considered to be the consequence of a phenomenon called epigenetic inheritance. The term “epigenetics” was initially coined by Conrad Waddington [2], and the concept that epigenetic changes are heritable across generations was subsequently introduced by Robin Holliday [3,4]. The idea that epigenetic inheritance is affected by environmental factors in the absence of DNA mutagenesis and that the resulting changes could be propagated to subsequent generations via the germline has been a matter of considerable debate (e.g., [5,6,7]). There is, however, rapidly accumulating evidence that DNA methylation and histone modifications are important for the regulation of numerous cellular functions and developmental processes across generations (reviewed in [8]).

Epigenetic mechanisms are critical for establishing and maintaining cell identity, whereby the developmental origin and the tissue microenvironment determine the actual epigenetic state at various genomic regions. It was demonstrated that cells derived from the same region within the ectoderm layer, such as keratinocytes and breast luminal and myoepithelial cells, share more genomic regions that possess similar epigenetic marks, than cells from the same neonatal skin that originate from a different region within the layer [9]. Somatic cell nuclear transfer and transcription factor-based reprogramming experiments can switch adult cells back to an embryonic state, thereby yielding pluripotent stem cells. It is noteworthy that the transcription factor-based reprogramming approach conserves the epigenetic memory of the tissue from which the cells originate [10]. Thus, cellular mechanisms guard the mechanisms of “origin of the cells”, and epigenetic features play an essential role in these mechanisms.

In this review, we summarize classical genetic and epigenetic concepts and then focus on molecular mechanisms underlying the recently discovered phenomenon of epigenetic transgenerational inheritance [11].

## 2. The DNA Code-Based Genotype-Phenotype Relationship

DNA encodes information needed for cells to grow, divide and differentiate and for tissues and organs to form and function in multi-cellular organisms. For this to work, DNA needs to be replicated and distributed during a process called mitosis. DNA replication is error-prone but still remarkably efficient because multiple mechanisms mitigate changes in the genetic code that may have negative consequences [12]. Since it was thought that the DNA’s genetic code determined the fate of cells and organisms, studies that examined the connection between the environment and disease heritability mostly focused on the relationship between specific exposures and DNA mutations in the germline [13,14]. Genome-wide association studies in human families helped identify single nucleotide polymorphisms (SNPs) that are associated with increased disease risk [15,16]. Complementary mouse gene deletion models were critical in linking mutations in a single gene with specific phenotypes, thereby explaining heritable traits [17]. These rodent studies strengthened the link between altered genotypes and human diseases. However, genetic studies failed to establish the mechanisms underlying the establishment of phenotypes not thought to involve changes in the DNA code; for example, morphological variations caused by environmental factors within inbred mouse strains [18], the striking morphological and physiological differences between bee workers and queens [19], or the connection between phenotypes and environmental factors [20,21].

During gametogenesis, DNA replication followed by two rounds of chromosome separation without an intervening S-phase ensures the maintenance of the diploid genotype through high fidelity copying of the genetic material [22]. However, the process of inheritance is not entirely indifferent to the genetic information it is transmitting. Part of the inheritance mechanism includes the faithful reconstitution of the heterogeneous parental states. For example, a particular methylation pattern or a chromatin configuration during a given developmental stage could either be permanently retained or erased and re-established at a subsequent stage [23]. The fluid nature of epigenetic marks as compared to relatively rigid DNA sequences makes epigenetic mechanisms better suited for rapidly responding to environmental stimuli. Modification of the epigenome, therefore, plays a critical role in expediently fine-tuning the flow of information encoded in the genome across multiple generations, without the need to permanently alter the underlying DNA sequences. This “soft inheritance” likely allows for an efficient adaptation to a rapidly changing environment and is therefore crucial for maintaining a species in its habitat.

## 3. Epigenetic Regulation via DNA Methylation

The notion of how non-genetic information could be recorded in the form of physical epigenetic marks started to take shape when modification of DNA’s cytosine base by methylation was discovered [24,25]. In the adult, approximately 25,000 protein-coding genes are expressed in different cell types from the same genome (with a few notable exceptions) due to mechanisms that are, by definition, epigenetic and are inherited during the mitotic cycle (reviewed in [26,27]). It is tempting to think that the mechanism active in preserving the epigenetic identity of differentiated cells can also work in the germline, to record and preserve environmentally induced epigenomic changes. Cytosine methylation provides a molecular basis for heritable genome-wide epigenetic marks in somatic cells that are faithfully reestablished after each round of DNA replication. This observation provides support for the hypothesis that similar DNA modifications might persist in the germline and affect gene expression in subsequent generations [4]. An extra level of regulation offers additional functionality for the DNA sequence, but is not without perils, since methylation errors could result in epigenetic defects in somatic cells and may lead to malignant cell transformation or premature aging.

Promoter sequences of mammalian genes can be classified into those with TATA boxes that initiate transcription at well-defined sites and those that contain CpG-rich stretches (islands) [28]. Although most of the DNA methylation in the genome occurs at CpG sites, methylated cytosine residues are also found at non-CpG sites. Non-CpG methylation has been found in neurons, pluripotent stem cells, and ovaries [29,30,31]. DNA methylation is catalyzed by canonical DNA methyltransferases (DNMTs) that include DNMT1, the maintenance methyltransferase, and DNMT3A/B, also known as de novo methyltransferase. The non-canonical DNA methyltransferases, DNMT2 and DNMT3L, do not possess catalytic activity, and their biological functions in humans remain elusive [32]. During DNA replication, DNMT1 plays an important role in maintaining the methylation pattern on the newly synthesized DNA strand, which is largely dependent on the methylation status of the template strand. DNMT1 localizes to newly synthesized DNA and is associated with the replication complex during S, G2, and M phases [33]. Importantly, global methylation levels were reported to be stable during replication and arrest [34]. Recently, a new DNA methylation enzyme encoded by *Dnmt3C* was described. The gene is a paralog of *Dnmt3B* created during an ancient genome duplication event. DNMT3C is very selective since it only appears to methylate the promoters of the young evolutionary retrotransposons; it is also active in fetal gonads [35].

It was suggested that exposure to toxic compounds causes changes in DNA methylation patterns. For example, exposure to low doses of uranium causes hypermethylation in testis and hypomethylation in ovaries with significant changes in the expression of DNA methyltransferase genes DNMT1 and DNMT3A/B [36]. The effects are maintained across F0, F1, and F2 generations [36]. The sensitivity of DNA methylation changes with age has also been reported: global DNA hypermethylation was found to be associated with high serum levels of persistent organic pollutants in an elderly population [37]. The effects of various pollutants on DNA methylation were summarized in a recent review [38]. The authors described the effects of a large number of pollutants, including Aflatoxin B1, arsenic, Bisphenol-A, metals, polycyclic aromatic hydrocarbons, tobacco smoke, persistent organic pollutants, and nutritional factors on DNA methylation. Different genomic regions are affected by each of these compounds, suggesting that they elicit their effects on specific targets. Remarkably, changes in the methylation state of retroelements could be caused by several compounds, including arsenic [39], cadmium [40], and air pollution [41]. In smokers, the DNA methylation state was altered in genes encoding detoxification enzymes, including AHRR, CYP1A1, CYP1B1, and CYTL1 [42]. Social factors such as stress also cause changes in DNA methylation. For example, maternal depression during the prenatal period is linked to altered methylation of brain-derived neurotrophic factor (BDNF) [43]. Maternal anxiety in pregnancy is associated with decreased *IGF2/H19* ICR DNA methylation in progeny at birth and with low birth weight in neonates [44]. Lifestyle factors, such as physical activity (reviewed in [45]), or health outcomes (obesity [46], diabetes mellitus [47]) can also influence DNA methylation. Thus, chemical, physical, nutritional, urbanistic, and lifestyle factors likely affect DNA methylation patterns. Taken together, human exposure data suggest DNA methylation to be highly affected by pollutants, which appears to trigger compound-specific consequences. A change in DNA methylation enzyme activities seems to be a major cause of DNA methylation changes.

## 4. Epigenetic Regulation and Chromatin Proteins

In eukaryotes, genomic DNA of somatic cells is tightly packaged by being wrapped around core histones. Histones are highly alkaline proteins that organize DNA into its principal structural units, called nucleosomes. Nucleosomes are condensed into chromatin that forms chromosomes. Dimers of histones H2A, H2B, H3, and H4 form a histone octamer, which binds and covers approximately 1.7 turns of DNA helix, which corresponds to 146 base pairs. In addition, there are noncanonical histone variants, which can replace canonical histones. During processes such as DNA replication, DNA damage and the activation of gene expression, noncanonical histones are incorporated into nucleosomes. The most common noncanonical variants are H3.3 and centromeric H3, also known as centromeric protein A (CENPA) in humans [48]. Histone H2A also exists in several variants. For example, the H2AX variant replaces histone H2A and becomes phosphorylated during the DNA damage response [49], chromatin remodeling, and X-chromosome inactivation in somatic cells [50]. H2A.Z regulates transcription, DNA repair, suppression of antisense RNA and RNA polymerase II recruitment [51]. H2A.B positively regulates transcription elongation by overcoming DNA methylation in the transcribed region [52]. There are also sex-specific variants of histones. For example, testicular histone H3.5 was found to play a role in DNA synthesis [53].

Most nucleosomes also recruit either histone H1 or high mobility group (HMG) proteins to form a particle known as the chromatosome; reviewed in [1]. Histone H1 functions as a so-called linker histone. It binds to the nucleosome units to form higher-order structures; reviewed in [54]. Linker histones have large numbers of post-translational modifications (PTMs) and are important for the regulation of DNA replication, DNA repair and genome stability; reviewed in [54]. Linker histones include 12 subtypes that are expressed in humans and mice. Individual subtypes are organized in three distinct groups: the somatic replication-dependent subtypes (H1.1–H1.5), the somatic replication-independent variants (H1.0 and H1.10) and the germline-specific subtypes (H1.6 (TS), H1.7 (TS), splice variants (H1.8 (OO) and H1.9 (TS)) [55]. Mammalian oocytes-specific histone, H1foo, was found to be present until the embryonic two-cell stage. It is thought that H1foo prevents the differentiation of mouse embryonic stem cells by maintaining the expression of pluripotent genes through the regulation of the chromatin structure [56].

Actively transcribed regions in the genome possess an open (accessible) chromatin state. These accessible domains contain DNase I-sensitive regions, which cover large upstream and downstream regions of the transcription units they are associated with [57]. However, some active genes remain in a condensed state where the linker regions are protected, such that they are no longer accessible to DNase I [58].

The histone tails in canonical, noncanonical, and linker histones are frequently modified. The PTMs include, among others, methylation, acetylation, phosphorylation, glycosylation and ubiquitination; for review, see [59] and [60]. The most common PTMs are methylation and acetylation of lysine (K) residues in the N-termini of histone 3 (H3) and histone 4 (H4). Typically, acetylation is associated with transcription activation, whereas de-acetylation often induces transcription repression. Histone methylation is associated either with activation or repression of gene expression, depending on the location of the modified lysine in the protein [60].

DNA organization in gametes is notably different from that of the soma and harbors specialized chromatin patterns that enhance the unique function of the germline [61]. To control their DNA function, germ cells also utilize several specialized regulatory pathways, either by expressing specific transcription factors or unique histone isoforms. Histone modifications are believed to be especially important during spermatogenesis, where expression of histone variants follows a highly orchestrated order. Furthermore, developing sperm cells undergo a process of dedicated chromatin remodeling, the histone-to-protamine transition, that requires specialized transient (transition) proteins that condense chromatin in a way unique for the male germline. As discussed below, histone modifications and organization are likely to be one of the major contributors to transgenerational epigenetic inheritance.

It was suggested that histone PTMs involved in chromosome silencing and chromatin compaction persist at least through several rounds of mitotic divisions; reviewed in [62]. In addition to DNA replication reassembly, other mechanisms exist that can facilitate a “histone memory” transfer. For example, epigenetic transmission across various stages of spermatogenesis likely involves a combined effort of replication-dependent factors such as anti-silencing factor 1 (ASF1), chromatin assembly factor 1 (CAF-1), proliferating cell nuclear antigen (PCNA) and other chaperones that control histone PTMs; reviewed in [62].

Histone methylation at lysine 4 (H3K4me3) is the most studied PTM. This mark is associated with transcription and regions with open chromatin [63]. H3K4me3 plays a unique role across all developmental stages in mammalian species. In the early embryo, the paternal histones have low but detectable levels of H3K4me3 marks [64,65]. The mark is preserved in genes important for reprogramming, meiosis and spermiogenesis [66]. Finally, this mark, together with H3K27me3, which is established by the Polycomb repressor complex, is enriched at many promoters in ES cells [67]. It has been postulated that H3K27me3 and PRC2 transmit the state of repression across generations in *C. elegans* [68]. The PRC2 complex is composed of four subunits: the SET-domain-containing histone methyltransferases enhancer of zeste (EZH1 or EZH2), embryonic ectoderm development (EED), suppressor of zeste (SUZ12), and the CAF1 histone-binding proteins RBBP4 and RBBP7; for review, see [69]. Vertebrate studies also suggested the possibility that PRC and their antagonists (TrX proteins) are involved in transgenerational inheritance [70,71]; reviewed in [69]. A schematic diagram explaining the transgenerational inheritance of histone modifications is shown in Figure 1.

Besides methylation at lysine, histones can be methylated at arginine positions. Histone arginine methylation is involved in the regulation of signal transduction, transcription, RNA splicing and transport. Deregulation of protein arginine methyltransferases (PRMTs) is associated with a poor recovery prognosis in many cancers; reviewed in [72]. PRMT1 regulates the telomere length and stability since the depletion of PRMT1 leads to telomere doublets and promotes telomere shortening [73]. Given that PRMT1 is expressed during development [74] and PRMT5 is important for germ cell specification [75], it is conceivable that arginine methylation is also implicated in transgenerational epigenetic inheritance (TEI).

In *C. elegans*, the transgenerational inheritance of temperature-induced changes in the expression of heterochromatic genes was associated with altered trimethylation of H3K9 over fourteen generations [76]. In addition, H3K9me3 is critical for establishing heterochromatin [77] and essential for normal meiosis [78]. Gestational exposure to bisphenol decreased H3K9me2 and H3K9me3 in germ cells of the neonatal testis [79], suggesting a possible role for these marks during early developmental stages.

In many species, the crosstalk between histone modifications is important for establishing the chromatin state. For example, pharmacological acetylation using HDAC inhibitors, such as valproic acid (VPA), triggers replication-independent active demethylation at a global level [38]. VPA can alter the state of methylation of genes involved in tumor growth and metastasis [80]. A functional connection between histone H3 methylation and DNA methylation was also shown [81,82]. One excellent example of the crosstalk between different mechanisms is the transition of primordial germ cells to gonocyte fate. In germ cells, the reprogramming process involves the coordinated interplay between promoter sequences, DNA methylation, PRC1, and DNA methylation-dependent- and independent function of TET1. A critical set of germline reprogramming responsive (GRR) genes are involved in gamete generation and meiosis, and these genes possess specific histone marks and binding sites for transcription factors [83]. For more information, we refer the readers to an excellent review of work in this field [84]. In conclusion, changes in histone modifications are essential for establishing the chromatin state at different developmental stages.

## 5. Epigenetic Control Mediated by Short and Long Non-Coding RNAs

Non-coding RNAs (ncRNAs) are defined as transcripts that are not translated into biologically active proteins. Classic examples are ribosomal RNAs (rRNAs) and transfer RNAs (tRNAs) required for ribosome biogenesis and protein translation. Recent advances in genome-wide RNA profiling technologies substantially increased the number and types of non-coding RNAs known. They are broadly and somewhat arbitrarily organized into small (<200 nucleotides) and long (>200nt) ncRNAs [85]. The former class includes small interfering RNAs (siRNAs), micro RNAs (miRNAs), piwi-interacting RNAs (piRNAs), small nucleolar RNAs (snoRNAs), small nuclear RNAs (snRNAs), and tRNA-derived fragments (tRFs) [86]. Among these, siRNAs, piRNAs, and miRNAs are known to have a regulatory role in epigenetics [87]. Short ncRNAs are known to be transcribed from H3K4me3-enriched PRC2 target genes that undergo cell-specific silencing [88]. They interact with PRC2 through a stem-loop structure resulting in gene repression and H3K27 methylation in *cis,* thereby stabilizing the complex at the site of transcription [88].

siRNAs are derived from double-stranded RNA precursors that are cleaved by DICER to form 19-24 base RNAs, that interact with argonaute proteins to repress mRNA translation [89]. siRNAs act via the RNA interference (RNAi) pathways to silence gene expression through DNA methylation and histone modifications [90,91]. In fission yeast, siRNA-mediated silencing pathways are involved in chromatin modifications and heterochromatin formation [92]. Deletion of *ago1* (argonaute), *dcr1* (dicer), and RNA-dependent RNA gene homologs that are key components of the RNA-induced transcriptional silencing complex (RITS) causes abnormal accumulation of long non-coding RNAs and loss of histone H3K9 methylation, resulting in impaired centromere function [93].

piRNAs fall into a large class of ncRNAs that interact with the piwi-type of argonaute proteins. A major role of these small transcripts (26–31 nt) for which many thousands of species were identified in fly, mouse, and human, is to transcriptionally silence retrotransposons in the germline. piRNAs are expressed in both male and female gametes and appear to localize both to the nucleus and the cytoplasm. Currently, they are only known to be important in male reproduction. Recent studies have shed light on the role of piRNAs in transgenerational inheritance; reviewed in [94]. In *C*. *elegans*, the interaction of PRG-1 (the main PIWI protein) with its cytoplasmic target results in the synthesis of 22G RNAs (a class of small RNAs) which, in turn, get loaded onto argonaute proteins including WAGO-9. This loading event facilitates the entry of WAGO-9 into the nucleus where it interacts with nuclear RNAi factors and triggers transcriptional silencing as well as trimethylation of H3K9 at the genomic loci that encode the target RNAs [94]. Remarkably, this silenced state gets stably transmitted across generations [95].

miRNAs (21–24 nt) also function in gene regulation by pairing to homologous regions within mRNAs, thereby forming double-stranded RNAs (dsRNA) that prevent the mRNA’s translation by inducing its degradation or preventing its efficient interaction with ribosomes during translation. miRNAs tend to be conserved, which underlines their important roles in the (more or less subtle) regulation of numerous protein’s cellular levels in mammals. It is noteworthy that miRNAs can interact with multiple (sometimes hundreds) of mRNAs via a small (6–8 nt) “seed region”, which enables a fairly limited number of known miRNAs to influence a substantial portion of the transcriptome and proteome. The influence of environmental factors on miRNAs has been documented. In cigarette smokers, 28 miRNAs were found to be differentially expressed in spermatozoa when compared to non-smokers [96]. In mice, the miRNAs in spermatozoa play a critical role in gene expression during embryogenesis and can induce altered phenotype in the progeny [97]; reviewed in [98]. Moreover, early life traumatic stress experiences in mice alter miRNA expression, which results in metabolic and behavioral changes in the progeny [99]. Therefore, the potential role of miRNAs in transmitting epigenetic information via transgenerational inheritance appears to be possible and therefore warrants further investigation.

The genomes of multicellular organisms encode a wide variety of long non-coding (lncRNAs), many of which show a preferential or even tissue-specific expression pattern [100]. They are typically transcribed and processed like mRNAs but unlike known protein-coding transcripts they are thought to have little, or no coding potential and the vast majority are not conserved during evolution [101,102]. Whether or not lncRNAs productively interact with ribosomes is a matter of debate. However, recently emerging evidence from model organisms such as budding yeast and work in mammals raises the fascinating possibility that a substantial fraction of transcripts currently annotated as lncRNAs may actually encode a wide range of small proteins and peptides [103,104]. Indeed, some of them appear to be stable in vivo, and a few cases are now known where very small proteins play important biological roles, notably in muscle tissues [105].

Regardless of their actual coding potential, it is well established that numerous lncRNAs play regulatory roles during cell growth, development and disease. They include epigenetic processes, notably via binding to DNA, chromatin factors, and regulatory proteins. Such interactions are mediated by modularly folded domains that mediate lncRNA-DNA, lncRNA-RNA, and lncRNA-protein binding [106]. The classic example of lncRNA-mediated chromatin regulation is a process called X-chromosome inactivation whereby one X chromosome is inactivated in females to ensure an equal dosage of X genes in males and females; the readers are referred to excellent reviews in the field [107,108]).

For ncRNAs to have an effect across generations, they must be physically transferred from the parents to the offspring via the germline. A large body of evidence demonstrates that male germ cells at mitotic, meiotic and post-meiotic stages and female gametes stably express thousands of ncRNAs [109,110,111]. An earlier RNA profiling study of human sperm led the authors to conclude that the male gamete is not only a vector of DNA but also appears to contribute to the initial transcriptome of the fertilized egg [112,113]. Such inherited lncRNAs could then participate in epigenetic regulation processes *in trans* across numerous loci or *in cis*, for example at selected maternally or paternally imprinted loci; reviewed in [114].

Another layer of complexity within transgenerational epigenetic inheritance mediated by RNA-dependent processes, is the emerging epitranscriptome. This term refers to molecular alterations in both mRNAs and ncRNAs that do not change their ribonucleotide sequence [115,116]. There is no doubt that this nascent field will contribute to a better understanding of how epigenetic traits are passed on across generations.

## 6. Germ Cell Specification and Reprograming

Although epigenetic patterns are relatively stable in somatic cells during adult life, the epigenome is reprogramed during development to acquire totipotency; reviewed in [117]. The global reprogramming events occur in germ cells during embryogenesis. In contrast, in differentiated cells such as spermatocytes or spermatids, epigenetic changes were also found during postnatal spermatogenesis. However, these cells still possess the epigenetic signature of the germ cell lineage they stem from.

Epigenetic inheritance thus has to be reconciled with reprogramming, a key process essential for organogenesis and germ cell production. To give rise to a totipotent zygote able to generate an enormous variety of cell types in the body, any sex-specific epigenetic programs in the germline have to be erased. In mammals, the first reprogramming of the epigenome occurs during early embryonic development from the zygote stage to the formation of layers and the second one occurs during the somatic-to-germline transition (E6.5–E13.5). In males, a third event occurs during spermatogenesis. Inadequate epigenetic reprogramming of the donor nucleus is thought to be the major reason for the developmental failure of cloned embryos. Germ cells of both sexes that fail to reprogram their genomes produce deficient offspring [118]. Genome reprogramming in the germline requires the transcription factor BLIMP1/PRDM1; reviewed in [119]. In mouse germ cells BLIMP1 cooperates with PRTM5 (a histone arginine methyltransferase) and the transcriptional repressor LSD1 to coordinate histone modifications [75,120]. BLIMP1 also acts together with PRDM14 and AP2γ to induce the expression of pluripotency genes and to suppress the expression of somatic genes in mice [121].

Human primordial germ cell-like cells (hPGCLCs) were derived from human embryonic stem (ES) cells in vitro [122]. It was suggested that due to obvious differences between mouse and human pluripotent ESCs, human hPGCLCs do not use similar mechanisms in their specification as mouse cells [123,124]. In humans, SOX17 is a factor involved in the human primordial germ cell (hPGC) specification and is a key regulator of their cell fate [125]. BLIMP1 is a downstream factor of SOX17, and it represses the endodermal and somatic genes during hPGCLC specification [125]. FGF2 and TGF-β cytokines have divergent functions in promoting pluripotency in humans. They induce pluripotent KLF4 and NANOG transcription factor expression in human ESCs, but not in murine epiblast stem cells (EpiSCs). In the mouse, they regulate pluripotency priming [126]. This suggests that the effects on FGF2 and TGF-β cytokine pathways could also affect germ cell specification. This resetting process and its fidelity assure that the new lineages are well established and functional. It was found that GRR genes could trigger the program of gonocytes in embryos. The epigenetic state at GRR genes is established by the coordinated action of transcription factors and epigenetic modifier enzymes. Notably, some GRRs are involved in meiosis and gametogenesis, such as *Dazl*, *Sycp1-3*, *Rad51c* [83]. Thus, somatic-to-germline reprogramming is a comprehensive process underlying germline development, and any perturbation of this process likely affects not only current but also future generations.

For epigenetic inheritance to occur, the pattern of chromatin modifications altered in the germline during an individual’s lifetime has to successfully pass or evade the control mechanisms that preserve the integrity and totipotency of the germline’s genome. The systematic resetting of epigenetic marks between generations represents the largest hurdle to conceptualizing epigenetic inheritance. During spermatogenesis, genome-wide sex-specific remodeling of the epigenome occurs as a highly orchestrated process of DNA demethylation, followed by DNA re-methylation and chromatin modification.

## 7. Retrotransposons Successfully Evade the Second Reprogramming Wave

Intracisternal A particles (IAPs) are one of the best-studied DNA elements that can mediate transgenerational inheritance. IAPs belong to the endogenous retrovirus (ERVs) family. ERVs and other repetitive DNA elements are an integral part of the mammalian genome. According to a recent estimate transposable elements may constitute up to two-thirds of the human genome [127]. ERVs are characterized by multiple copies of retro-elements carrying long terminal repeats (LTRs). Most mammalian ERVs are inactive and do not impair the genome integrity. However, a subset of these DNA elements retains their inherent transposition activity that differs quite dramatically between mammalian species [128]. Mice are prone to carrying numerous active LTR elements and insertional mutagenesis by IAPs is particularly effective in these species [129]. Approximately 10% of spontaneous mutations in mice result from ectopic ERV insertions and are associated with cancers [130].

Mammalian cells developed effective mechanisms to suppress residual ERV transposition activity. Most importantly, DNA methylation of ERV sequences is critical for their suppression in post-implantation embryos and the male germline; reviewed in [131]. During early preimplantation and primordial germ cell (PGC) development mammalian genomes undergo genome-wide DNA demethylation [132]. However, IAPs, an especially mutagenic class of ERVs, +are impervious to DNA demethylation and mostly preserve their methylated state [133].

Other, less active ERVs (e.g., MuERV-L and ETn/MusD) are silenced through epigenetic preservation of their inactive chromatin configurations. Specifically, preserving methylation at H3K9 was shown to be important, and inactivation of the H3K9me3 histone methyltransferase SETDB1 and its co-repressor TRIM28 led to re-activation of ERVs in PGCs [134,135]. In ES cells, DNA methylation and H3K9me3 regulate largely non-overlapping subsets of ERVs, with the notable exception of IAPs whose silencing appears to require the synergistic action of both epigenetic marks [136]. This synergy is accomplished by the activity of the G9a/GLP complex that recruits DNA methyltransferases to preserve IAPs methylation [137]. In a complementary fashion, IAPs’ H3K9me2-enriched regions are refractory to demethylation via recruitment of the DNMT1 chaperone NP95/UHRF1 [138,139].

The second wave of global DNA demethylation occurs after fertilization in the inner cell mass compartment of the embryo. Methylation patterns associated with cellular differentiation into developing embryonic lineages are then re-established *de novo* in the blastocyst. IAPs resist the second wave of reprogramming, and as a consequence, genes with IAP elements located nearby fail to be reprogrammed even in PGCs [140]. At E13.5 in the mouse, when developing PGCs are extremely hypomethylated, most IAPs methylation marks persist and CpG islands near IAPs remain methylated. Additional partially erased CpG islands were also observed, implicating these variably erased sequences as carriers of transgenerational epigenetic inheritance [141].

## 8. Resetting of the Parental Epigenetic Makeup in Early Zygotes Reveals the Epigenome’s Plasticity

It has been known for more than 40 years that histones are not completely displaced by protamines in sperm and a specific developmental function for the histone-enriched part of the chromatin was postulated [142]. However, whether a specific portion of the sperm chromatin is consistently enriched in histones remained unconfirmed until recently. The alternative possibility to consider was residual histone presence due to imperfect protamine replacement, leading to a random distribution of nucleosomes that lacks functional significance. Recent global profiling of the histone modifications in human and mouse spermatozoa has indicated that ∼4% of the haploid genome in humans and ∼1%–2% in mice consistently retain their nucleosomes in mature sperm. Importantly, the authors showed that certain genes preferentially maintain histone H3 lysine 27 trimethylation (H3K27me3) marks at their promoters [143,144]. This raises the possibility that histone marks can be strategically deployed to propagate epigenetic information across generations.

Recently improved chromatin immunoprecipitation and sequencing (ChIP-Seq) assays that can derive reliable data from a limited number of cells from early embryonic stages, shed new light on the fate of parental histones. Upon fertilization, H3K4me3 marks are depleted in zygotes and only observed after major zygotic genome activation (ZGA) at the late two-cell stage. However, the enrichment of paternal H3K4me3 is still weaker than the maternal one, even in the ICM. Paternal H3K4me3 showed enrichment in zygotes, suggesting the possibility that these marks may be poised in some promoters for late activation and could transmit a paternal epigenetic memory [145]. Overall, these data are consistent with the notion that histone marks are newly acquired after fertilization via a mechanism that remains to be determined.

## 9. Epimutations Can Have Long-Term Consequences in Determining Phenotypes

Epigenetic inheritance has been found in all taxonomic groups and is therefore likely to be ubiquitous. However, the fidelity with which epigenetic states are transmitted is variable. Alterations in epigenetic states are termed epimutations. Our understanding of the rates and causes of epimutations remains rudimentary. The existence of heritable epigenetic variations that could be developmentally and environmentally re-programmed is hard to reconcile with a strictly interpreted mutation-driven concept for evolution. Surprisingly, while epigenetic inheritance limited to the developmental regulation of specific genes *via*, for example, imprinting is now widely accepted, the transgenerational inheritance of epigenetic marks altered by environmental factors remains controversial.

One of the strongest pieces of evidence for heritable chromatin states in gametes comes from studies in *C. elegans*. Mutant worms that lack *spr-5*, a H3K4 lysine-specific histone demethylase 1 (LSD1/KDM1), become progressively sterile over the course of ~20 generations and display increased H3K4me2 levels in later generations and dysregulation of spermatogenesis genes [146,147]. *spr-5* mutants also gradually decrease their brood sizes and their progeny becomes infertile at the 20th generation. A reintroduction of a single wild-type *spr-5* allele restores reproductive function even in late (severely sterile) generations, demonstrating that *spr-5* is both necessary and sufficient for epigenetic resetting. This work shows that H3K4me2 is an important component of the regulatory circuitry that establishes epigenetic memory in the worm and that *spr-5*-mediated erasure of this mark is essential for appropriate germline transmission of epigenetic patterns from one generation to the next [146,147]. Similarly, in flies, responses to toxic insults were epigenetically inherited in subsequent generations of unexposed offspring. This response was partially mediated through the suppression of another H3 histone modifier belonging to the polycomb group genes [148]. In summary, these findings strongly implicate chromatin proteins in transgenerational epigenetic inheritance.

## 10. Environmentally Induced Epigenetic Mutations

Several environmental factors, including diet, can affect DNA methylation patterns in subsequent generations. *In utero* exposure to a high-fat diet induces paternal obesity and insulin resistance accompanied by changes in sperm micro RNA content and germ cell methylation in two generations of offspring [149]. Intrauterine hyperglycemia exposure contributes to intergenerational metabolic changes in the F2 but not the F3 generation [150].

The abundant use of industrial and household chemicals causes widespread concerns about their long-term impact on animal and human health. Chemical compounds that are either known or thought to perturb endocrine signaling are called endocrine-disrupting chemicals (EDCs). These compounds are ubiquitous in the environment since they are major components of numerous pesticides and they are present in commonly used products, such as food, food packaging materials and cosmetics. According to the World Health Organization’s report from 2017, poorly regulated pesticide use can negatively affect humans and wildlife around the world [151]. There is an urgent need to better understand the long-term consequences that EDCs have on human health and to distinguish the benign outcomes from the potentially harmful ones.

In any rodent study that aims at revealing a transgenerational effect, the third generation (F3) must be analyzed because it is the first generation not directly exposed to the compound in question. This is the case because pregnant animals (F0) are exposed to toxicants during embryonic days E6-E15 (corresponding to the epigenetic reprogramming stage). Here, the fetus, which is the F1 generation, is directly exposed *in utero*. After birth, F1 males are mated with unrelated and non-littermate females to derive the F2 generation. Since the F2 generation is derived from the gametes of F1 that were exposed *in utero*, F3 is the first non-exposed generation. F3 is then generated by mating F2 males with unrelated and non-littermate females. In Table 1 we summarize a few recently published examples of transgenerational effects by environmental factors on epigenetic marks in rodent species. Several studies of different chemical compounds by the Skinner laboratory using different chemical compounds showed that exposure during a critical reprogramming window in F0 leads to transgenerational effects in F3 and F4 generations [152,153,154,155,156,157]. Critically, these studies revealed that DNA methylation in sperm was altered. In related work, F3 offspring of female mice exposed to tributyltin (TBT) throughout pregnancy and lactation, was predisposed to obesity due to altered chromatin organization that subsequently biased DNA methylation and gene expression [158]. In our recent studies, we showed that exposure to the herbicide atrazine or the pesticide chlordecone promotes transgenerational effects and the changes were associated with altered histone trimethylation at lysine 4 [159,160].

For how long are these induced changes maintained? Results reported in the literature currently provide no definitive answer to this question. In some cases environmentally induced epigenetic changes are clearly temporary. Such a phenomenon is comparable to the gradual restoration of heterochromatic regions perturbed by environmental stress: the “healing” of an “epigenetic wound”. In other words, the capacity of an organism to restore disturbed heterochromatin states within a single generation can be limited but eventually the “wound” will heal. Another aspect to consider, even in the absence of continuous exposure, is that the environmental situation may be beneficial for the altered phenotype and such (natural or artificial) selection can slow down the process and thus allow for the TEI’s maintenace in subsequent generations.

The most likely scenario is, however, that epigenetic effects are reversed gradually in each generation if the exposure that causes them is not continuous. For example, H3K4me3 peaks at the TSSs of *Klf1* and *Pou5f1* are less different in F3 than F1 generations [160]. We think that persistent effects over several generations occur due to changes in the level of expression of master regulator genes, such as the key pluripotency gene *Pou5f1* that could contribute to propagating the epigenetic effects. POU5F1 could directly alter the expression of up to 400 genes, which in turn would modulate a large number of down-stream target genes, ultimately affecting the global transcriptional network. Epigenetic memory could also be preserved through reprogramming events due to coordinated actions of several factors, including transcriptional factor TET1 [83].

Humans are generally sensitive to endocrine disruption during early childhood and puberty, but especially prenatally, during rapid development *in utero* [161]. There is evidence that exposure to EDCs during development is associated with low birth weight and premature birth [162]; reviewed in [163]. Some data also link EDCs with attention deficit disorders, autism [164], cryptorchidism and hypospadias [165,166]. While epidemiological studies correlate EDC exposure and human diseases, animal research has yet to demonstrate a firm mechanistic connection between the chemicals, their dosage and directly related pathologies. Prenatal and early exposure to diethylstilbestrol (DES), a non-steroid estrogen, induces developmental anomalies of the female reproductive tract. It is associated with permanent promoter hypermethylation and reduced expression of the homeobox HOX10A gene that regulates the organogenesis of the offspring [167]. In humans, *in utero* exposure to DES causes carcinomas in the uterus and reproductive issues in the daughters [168,169]. As previously mentioned, cigarette smoke induces specific differences in the spermatozoal microRNA content of human smokers compared with non-smokers, which is relevant for reproductive health because these altered microRNAs appear to predominantly mediate pathways vital for normal embryo and sperm development [96].

The nuclear receptors ER, AR and PPARy were suggested to be involved in triggering transgenerational inheritance via affecting DNA methylation; reviewed in [170]. For example, vincozolin causes transgenerational inheritance via AR signaling [152]. ER signaling is involved in the transgenerational inheritance of BPA [171,172,173,174], and the effects are mediated *via* DNA methylation [175]. Perinatal exposure to smoke causes downregulation of PPARy in lung fibroblasts [176] and mediates transgenerational effects in rat lungs [177]. On the other hand, nuclear receptors could mediate TEI via histone modifications changes. For example, overexpression of histone demethylase enzyme KDM1A in mice [70] or a knock-out of rbr-2 in *C. elegans* [178] leads to transgenerational morphological changes suggesting that histone-modifying enzymes are major players in TEI. The activities of histone modifiers could be regulated by nuclear receptors, such as ESR1. This protein is known to directly regulate the methyltransferase KMT2D, SUV39H1, and EZH2 [179]. We believe that environmental toxicants could directly or indirectly modulate nuclear receptors. For example, BPA can directly bind to ESR1 [180] and ERR-gamma [181] receptors. Thus, histone-modifying enzymes and nuclear receptors appear to be some of the major mediators of TEI.

Do epigenetic changes occur at random? Our data suggest that environmental toxicants induce changes in specific regions of the genome. For example, in two of our studies we found no overlap between altered regions. In ATZ-induced TEI we found an enrichment in binding motifs recognized by WT1 (Wilm’s tumor 1) and SP1/3/4 (specificity proteins 1, 3, and 4). In contrast, in CD-induced TEI we found a significant increase in ESR1 binding sites in F1 and F3 generations, suggesting that toxicants affect different sets of transcriptional factor binding motifs. Besides, experimental evidence of ESR1 binding at its targets in embryonic CD-exposed testes suggests that at least some regions in the genome could be specifically altered by CD, which is known for its ESR1 binding property [182].

At present, the underlying transgenerational epigenetic mechanisms of EDC-induced effects remain largely unknown. This could due to the fact that transgenerational studies are difficult to carry out. Box 1 describes the challenges that transgenerational epigenetic studies represent. Further mechanistic studies are needed to evaluate the deleterious effect of EDCs and to produce a critical mass of data for governmental authorities to regulate market access for compounds that may alter heritable epigenetic marks.

Box 1Challenges associated with transgenerational studies.**Outbred models**: Inbred strains are the best models to have an unbiased view of genetic and epigenetic variations following exposure to environmental factors. However, inbred strains are less susceptible to epigenetic transgenerational changes, and they do not reflect the complexity of the human genome. At the moment, most data are available on outbred strains.**Biological replicates and statistical analyses**: This is associated with the previous one. A large number of biological are required due to the high heterogeneity of outbred strains, and this is especially challenging when conducting studies up to three generations.**Short exposure window**: The developmental period is particularly susceptible to environmental factors due to epigenetic reprogramming, where the somatic-to-germline transition takes place. In mouse, this window is short, between E6–E15, and cannot be extended to avoid secondary effects associated with general toxicity.**Choice of the molecule**: For transgenerational studies, the environmental chemicals that are globally used or the ones that are of high-priority could be tested.**Establishing dose-response**: For studies involving environmental chemicals, the doses that induce transgenerational response without causing systemic toxicity needs to be experimentally established. Ideally, the minimal dose relevant to environmental doses where phenotypic effects are detected needs to be identified.**Sex-specific transgenerational effects**: Epidemiological studies have shown an association between various environmental factors and phenotypic changes in the offspring. However, it is difficult to study the exact mechanisms involved in maternal epigenetic inheritance due to the limited number of oocytes available in rodent models. Most studies that analyzed the epigenetic mechanisms in rodents to date focused on paternal inheritance.

## 11. Perspectives

In spite of considerable progress in understanding mechanisms underlying the transgenerational inheritance of epigenetic marks, there is still limited knowledge about the factors that contribute to the phenomenon. Only a few reports available in the scientific literature associate the effects of endocrine-disrupting chemicals with epigenetic transgenerational inheritance. Since most of the available studies were performed on animals it is still unclear whether similar mechanisms of action occur during the process of transgenerational inheritance in humans. The conservation of epigenetic mechanisms between humans and rodents is only beginning to be explored, and some important differences have been documented (for example, X chromosome inactivation mechanisms; reviewed in [183]). We suspect that the complexity of humans may lead to large inter-individual variations in response to toxicants, which will complicate direct interspecies comparisons. Ethical issues limiting access to human samples impair work on the processes that underlie TEI. We note, however, that conserved histone-containing fractions of the sperm genome in mammalian species suggest that at least some of the mechanisms regulating paternal epigenetic information transfer across generations might be conserved. The sex-specific components involved in transgenerational inheritance are currently not well understood. Here, the recent discovery of topologically associated genome domains, three-dimensional interactions of distally located transcriptional regulatory elements and novel linker histones contributing to chromatin organization open promising new avenues for efforts to gain further insight into transgenerational epigenetic inheritance.

## Figures and Tables

**Figure 1 cells-08-01559-f001:**
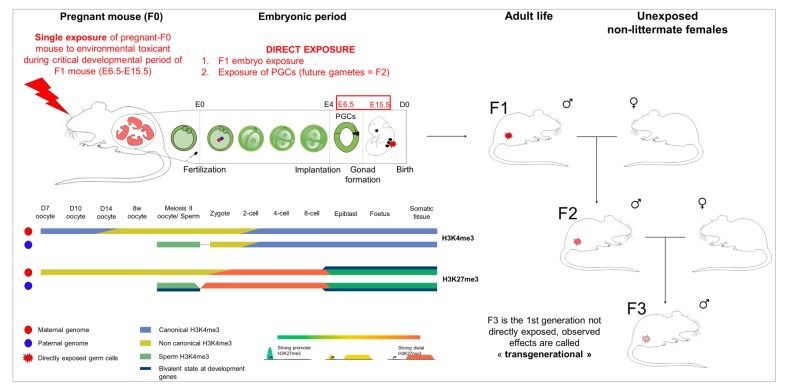
The transgenerational inheritance and histone modifications in the mouse. In the zygote, the paternal genome appears to be generally depleted of H3K4me3. Strong paternal H3K4me3 peaks reappear during the late two-cell stage. However, the levels of H3K4me3 become comparable between maternal and paternal genomes only after implantation. At the late two-cell stage, non-canonical H3K4me3 peaks (broad and low-intensity peaks) are replaced by canonical H3K4me3 (narrow and strong-intensity peaks). In early-stage (pre-implantation), H3K27me3 marks are enriched at distal promoters far from transcription start sites (TSS) in paternal and maternal genomes. At later stages (E5.5), they are generally enriched at TSS. Primordial germ cells move into the genital ridges at E11.5 and differentiate into spermatocytes or oocytes. Both canonical and noncanonical H3K27me3 are found in developing and mature oocytes. During spermatogenesis, histones are replaced by protamines, whereby only ~10% of histones are preserved. The environmental exposure to toxicants could promote changes in germline cells at any developmental stage, with more dramatic effects being observed during embryonic germ cell reprogramming. The exposed germline cells produce spermatozoa with altered epigenetic features. Finally, these epigenetic changes could be preserved up to several generations via histone retention mechanism. Figure adapted from [65,145,181].

**Table 1 cells-08-01559-t001:** Experimental evidence on the effect of paternal or maternal environmental factors on epigenetic changes in the offspring.

	Epigenetic Mark that Was Affected	Environmental Factor Involved	Organ/Matrices Studied	Animal Model	Associated Health Issue in Offspring	Reference
**Metabolic effects**	Hypomethylation of *Il13ra2* (interleukin 13 receptor subunit alpha 2), altered expression of 642 pancreatic islet genes in female F1 offspring following paternal high-fat diet.	Paternal high-fat diet	Pancreatic islets	Sprague–Dawley rats	Impaired glucose-insulin homeostatsis (Type 2 diabetes)	[184]
	20% change in cytosine methylation along with methylation at the enhancer of a key lipid regulator, PPARα, in the liver of F1 offspring following paternal low-protein diet.	Paternal low-protein diet	Liver	C57/Bl6 mice	Impaired cholesterol and lipid metabolism	[185]
	Altered epigenetic signatures in the insulin-2 gene promoter region and inefficient binding of transcription factor PDX1 at the insulin-2 promotor region following undernutrition for 50 generations.	Undernutrition (protein and caloric restriction)	Pancreas	Wistar rats	Adiposity/Type 2 diabetes	[186]
	Decrease in acetyl H3K9 and increase in dimethyl H3K9 levels in adiponectin and leptin gene promotor region in the offspring when mothers were fed high-fat diet for multiple generations.	Maternal high-fat diet	Adipose tissue	ICR outbred mice	Impaired glucose homeostasis and obesity	[187]
	Differentially methylated genes enriched for obesity/diabetic and metabolic changes in F2, not F3 generation males on intrauterine exposure to hyperglycemia.	Intrauterine hyperglycemia	Primordial germ cells	ICR mice	Obesity and insulin resistance	[150]
	Paternal diet restriction significantly changed the DNA methyltransferase, Dnmt1 and the transcript of methyl CpG binding protein 2 (Mecp2) in F1, not in F2 and F3 generations. An increase in the expression of histone modification gene, histone deacetylase 1 (*Hdac1*) in fetus liver was found in F1 and F2. An increased H3 acetylation in fetuses was also detected in F2 generation.	Diet restriction	Liver, adipose and muscle	Wistar rats	Metabolic changes	[188]
**Neurological effects**	Hypo- and hyper-methylation of several candidate genes including *MeCP2,* cannabinoid receptor 1 (*CB1*)*,* corticotropin-releasing hormone 2 (*CRFR2*) genes in the sperm of males and the brain of F2 females.	Stress (chronic maternal separation)	Brain, Sperm	C57Bl6/J mice	Depressive-like behavior	[189]
	An increase in DNA methyltransferase 1 (DNMT1) and ten-eleven translocation hydroxylases (TET1) in the frontal cortex and hippocampus of offspring along with a decrease in 5-methylcytosine and 5-hydroxylmethylcytosine levels at *Bdnf* gene regulatory regions following prenatal stress.	Stress (restraining movement in pregnant dams for 45 min from gestation day 7 until delivery)	Brain	Swiss albino mice	Schizophrenia-like phenotype	[190]
	Upregulation of miR-103 and downregulation of its target gene *Ptplb*, downregulation of mIR-145 (a marker of multiple sclerosis), upregulation of miR-323, miR-98 (involved in inflammatory responses in brain), and miR-219 that targets the gene *Dazap1* (marker of schizophrenia and bipolar disorders) was observed in the offspring following induction of stress to pregnant mothers.	Stress (Pregnant dams were forced to swim for 5 min and restrained body movement for 20 min from gestational day 12 to 18)	Brain	Long-Evans rats	Brain diseases (genes involved in multiple sclerosis, schizophrenia and bipolar disorder)	[99,191]
	CpG hypomethylation in the olfactory *Olfr151* gene in the sperm of F1 generation whose parents underwent olfactory fear conditioning with acetophenone.	Olfactory fear conditioning with acetophenone or propanol	Sperm	C57BL/6J and M71-LacZ transgenic mice	Fear/behavioral sensitivity	[192]
	Acetylation of H4K5 and H3K14, dimethylation of H3K4, and trimethylation of H3K36 (H3K36me3) were significantly decreased in mineralocorticoid receptor (*MR*) gene in the F2 hippocampus following maternal separation in F0 generation offspring. Sperm DNA methylation was significantly increased at several CpGs across the *MR* promoter. In a follow-up study, miR-375 was found to be upregulated in the hippocampus of F1 and F2 offspring.	Maternal separation and maternal stress for 2 weeks (F0)	Hippocampus Sperm	C57BL/6 mice	Traumatic stress/depressive anxiety-like behavior	[193]
**Cardio-vascular disorders**	Decreased methylation of the *AT1b* angiotensin receptor gene in the offspring following maternal low protein diet.	Maternal low protein diet	Adrenal gland	Wistar rats	Hypertension (renin-angiotensin system)	[194]
**Respiratory diseases**	Altered methylation of 14480 individual CpG loci in F1, 9413 loci in F2 and 6239 in F3 generations in dendritic cell methylome following maternal exposure to intranasal instillation of environmental particles.	Maternal intranasal instillation of environmental particles	Dendrite cells	BALB/C mice	Asthma	[195]
**Environmental toxicants/factors**	Transgenerational differential expression of 92 genes in the hippocampus and 276 genes in amygdala in males, and 1301 genes in hippocampus and 172 in the amygdala in females following exposure to vinclozolin, an endocrine-disrupting chemical.	Maternal exposure to vinclozolin (100 mg/kg/day from gestational day 8–14)	Brain	Sprague–Dawley rats	Anxiety-like behavior	[196]
	The number of methylated CpG in *H19* and *Gtl2* genes (paternally methylated) decreased while *Peg1*, *Snrpn,* and Peg3 (maternally methylated) increased in F1 male offspring. These effects were not significant in F2 and F3 generations.	Maternal exposure to vinclozolin (intraperitoneal injection at a dose of 50 mg/kg/day) from gestation day 10–18	Sperm	FVB/N mice	Decrease in sperm concentration	[197]
	Lower levels of microRNA, miR-130a, and increased levels of miR-16 and miR-221 along with a decreased expression of HIF-1α and other biochemical and histological changes in the lungs (F1 and F2 generations) where mothers were exposed to second-hand cigarette smoke (mice).	Maternal exposure to second-hand cigarette smoke	Lungs	BALB/C mice	Asthma and Bronchopulmonary dysplasia	[198]
	A significant increase in DNA methylation regions in sperm (also called epimutations) was observed in F1, F2, and F3 generations when pregnant rats were administered atrazine, a herbicide.	Maternal exposure to atrazine (intraperitoneal injection at a dose of 25 mg/kg body weight/day) from gestational day 8 to 14.	Sperm	Sprague–Dawley rats	Lean phenotype and hyperactivity	[199]
	Differentially methylated DNA methylation regions in sperm (epimutations) in 197 different promoters in the F3 generation were observed following the administration of a plastic compound mixture to pregnant rats.	Maternal exposure to plastic mixture (intraperitoneal injection of a mixture of bisphenol A 50 mg/kg BW/day, DEHP 750 mg/kg BW/day and DBP 66 mg/kg/BW/day) from gestational day 8 to 14.	Sperm	Sprague–Dawley rats	Obesity and sperm abnormalities	[155]

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
