# Peer review of "Transgenerational Inheritance of Environmentally Induced Epigenetic Alterations during Mammalian Development"

_cells, 2019, doi:10.3390/cells8121559_

Round 1

Reviewer 1 Report

This review organized by Legoff et al., is a really nice piece of work, and well written is in a grammatically good and very concise style. It brings the most updated information regarding the transgenerational inheritance of environmentally induced epigenetic alterations. This comes followed by some theoretical review and some of the most update experiments related to this field. Taking into account that this is not my main area of expertise, I profit very much from this review and thus, I believe it would also attract a considerable panel of readers. In my opinion, this could be published fast in its current form. I only make some observations that called my attention: 1) I missed the X-chromosome inactivation example since it’s one of the most notorious epigenetic examples. 2) The main observations are just correlations, but the real basis for this effect seems still missing. HOW some environmental factors affect epigenetic marks? Is that a random process? 3) “…the human genome’s complexity makes rather unlikely that mechanisms identified in rodents will ever fully explain the process in humans.” Do the authors really believe in that? Or due to generation time and other ethical factors, it’s just more difficult to identify them? 4) Another point that is not clear is how long such effects “resist” over generations. Are these epigenetic marks triggered by environmental factor reversible is the exposure is not continuous?

Congratulations on the nice work

Author Response

Reviewer 1

We thank the Referee for all the positive comments, and we address below the concerns raised in the critique.

1) I missed the X-chromosome inactivation example since it’s one of the most notorious epigenetic examples.

We agree with the Referee 1 that X-chromosome inactivation, imprinting etc. are the best-understood examples of mechanisms where epigenetics plays an important regulatory role. We have commented on this classical role of epigenetic factors and directed the readers to some excellent reviews in this area (Lines 320-323 in the revised manuscript).

2) The main observations are just correlations, but the real basis for this effect seems still missing. HOW some environmental factors affect epigenetic marks?

Several pieces of evidence suggest that changes in histone modifying enzymes are major triggers for transgenerational epigenetic inheritance (TEI). For example, overexpression of histone demethylase enzyme KDM1A in mice [1] or a knock-out of RBR-2 in C. elegans  [2] leads to transgenerational morphological changes suggesting that histone modifying enzymes are major players in TEI. The activities of histone modifiers could be regulated by nuclear receptors, such as ESR1. This protein is known to directly regulate the methyltransferase KMT2D, SUV39H1 and EZH2 [3]. We believe that environmental toxicants could directly or indirectly modulate the nuclear receptors. For example, BPA can directly bind to ESR1 [4] and ERR-gamma [5] receptors. Thus, histone modifying enzymes and nuclear receptors appear to be some of the major mediators of TEI (Lines 538-546 in the revised manuscript).

Is that a random process?

Our data suggest that environmental toxicants induce changes at specific regions at the genome. For example, in two of our studies we found no overlap between altered regions. In ATZ-induced TEI we found an enrichment in binding motifs recognized by WT1 (Wilm’s Tumor 1) and SP1/3/4 (Specificity Proteins 1, 3 and 4). In contrast, in CD-induced TEI we found significant increase in ESR1 binding sites in F1 and F3 generations, suggesting that toxicants affect different sets of transcriptional factor binding motifs. On the other hand, experimental evidence of ESR1 binding at its targets in embryonic CD-exposed testes suggests that at least some regions in the genome could be specifically altered by CD, which is known for its ESR1 binding property [6].  We included this statement in manuscript (Lines 547-555).

3) “…the human genome’s complexity makes rather unlikely that mechanisms identified in rodents will ever fully explain the process in humans.” Do the authors really believe in that? Or due to generation time and other ethical factors, it’s just more difficult to identify them?.

The conservation of epigenetic mechanisms between humans and rodents is only beginning to be explored, and some important differences have been documented (as in X chromosome inactivation mechanisms, reviewed in [7]. We suspect that the complexity of humans may lead to large inter-individual variations in response to toxicants, which will make drawing direct interspecies comparisons harder. The ethical issues and reduced accessibility to human material limits our understanding of the processes that underlie this phenomenon. We note, however, that similar histone-containing fractions of the sperm genome in mammalian species is consistent with the idea that at least some of the mechanisms regulating paternal epigenetic information transfer across generations are conserved. We included this statement in manuscript (Lines 570 to 579).

4) Another point that is not clear is how long such effects “resist” over generations. Are these epigenetic marks triggered by environmental factor reversible is the exposure is not continuous?

Literature suggests that there might be no definitive answer to this question. In some cases, environmentally induced epigenetic changes can clearly be temporary. Some compare it to the gradual restoration of heterochromatic regions perturbed by environmental stress, the ‘healing’ of an ‘epigenetic wound’. In other words, a capacity of an organism to restore disturbed heterochromatin states within a single generation can be limited, but eventually the “wound” will heal. Another aspect to consider, even in the absence of continuous exposure, environmental situation can remain beneficial for the altered phenotype and this (natural or artificial) selection can slow down the process and thus increase the possibility/duration of the TEI in subsequent generations.

The most likely scenario is the one where the epigenetic effects in each generation are reversed gradually if the exposure is not continuous. For example, the H3K4me3 peaks at TSS of Klf1 and Pou5f1 showed reduced difference in F3 compared to F1 generation [8]. We think that persistent effects over several generations occur due to changes in the level of expression of master regulator genes such as key pluripotency gene, Pou5f1 that could contribute to propagating the epigenetic effects. POU5F1 could directly alter the expression of up to 400 genes, that in turn would modulate the large number of other genes, and finally the global transcriptional network could be notably altered. Epigenetic memory could also be preserved through reprograming events due to coordinated actions of several factors, including transcriptional factor TET1 [9]. We included this statement in manuscript (Lines 498-515).

Reviewer 2 Report

The review article entitled "Transgenerational inheritance of environmentally induced epigenetic alterations during mammalian development" by Legoff et al. is a well-written review about environmentally induced epigenetic alterations and their inheritance through generations. This review is thorough and mostly updated all the recent literature about the phenomenon.  I would recommend publishing it in the current format with proofreading. 

Author Response

The review article entitled "Transgenerational inheritance of environmentally induced epigenetic alterations during mammalian development" by Legoff et al. is a well-written review about environmentally induced epigenetic alterations and their inheritance through generations. This review is thorough and mostly updated all the recent literature about the phenomenon. I would recommend publishing it in the current format with proofreading.

We thank the Reviewer for all the encouraging comments.

Reviewer 3 Report

Legoff et al present a review of epigenetic mechanisms involved in development and characterize transgenerational inheritance for environmental exposures. The following concerns about the manuscript should be addressed:

Page 2, lines 70-72: GWAS are not usually conducted in human families. Page 3, lines 123-125: This statement is missing a reference. Section 3 focuses on pollutants as environmental exposures that influence methylation, but we have learned from a number of EWAS studies that health outcomes themselves can influence methylation. The authors are missing a large sector of what types of environmental factors can influence methylation and do not acknowledge that other sources exist. Page 5, line 210: “see (Schuettengruber et al., 2017)” Figure 1: The current linear depiction of maternal and paternal genomes in this Figure seems to imply the same genomes are inherited through generations, when really these genomes are from different individuals. The arrows at the top also seem overly simplistic for transgenerational exposures and inheritance. An improvement to this figure may be to divide the generations in to separate panels that can more accurately reflect transgenerational exposures and methylation, as well as different genomes. “Multigenerational exposures” across most of the top of the figure also conflicts with “environmental toxicant exposure” only indicated to affect the fetus or testis gonad. Page 8, line 327: “Although epigenetic patterns are stable in somatic cells during adult life” is an inaccurate statement. Epigenetic patterns may be relatively stable compared to development, but there are epigenetic changes that can occur throughout life, which the authors previously describe through environmental exposures. Page 8, lines 333-334: “In males, a third event occurs during spermatogenesis.” What are the first and second events? Page 9, lines 354-363: There is not a clear message of this paragraph. Do the authors believe epigenetic marks are not passed through generations? The language in the last two sentences should be clarified and supported. What does “most epigenetic marks” and “their faithful preservation during global epigenetic reprogramming remains uncertain” mean? Page 9, line 400: The phrase “It has been known for a long time that” should be quantified or deleted. Figures 1 and 2: the generational labels within each figure seem to be in conflict with one another. Figure 2 does not convey a clear message without reading the legend, in which case only the text should be used rather than a figure. There is no depiction of the gametes that define F3 as the first unexposed generation and the red dashed arrows do not clearly show reduced epigenetic effects.

Author Response

While thanking the Reviewer 3 for all the valuable comments, we address below the questions raised.

Page 2, lines 70-72: GWAS are not usually conducted in human families.

We modified this sentence and removed word “families”.

Page 3, lines 123-125: This statement is missing a reference.

We have now added the missing reference (Reference 36).

Section 3 focuses on pollutants as environmental exposures that influence methylation, but we have learned from a number of EWAS studies that health outcomes themselves can influence methylation. The authors are missing a large sector of what types of environmental factors can influence methylation and do not acknowledge that other sources exist.

In Section 3 we briefly mentioned other environmental factors such as maternal anxiety and depression and their influence on DNA methylation. Considering the Reviewer’s suggestion, we have now expanded this list to include the role of other health outcomes/environmental factors such as obesity [10], diabetes [11] and physical activity, reviewed in [12]. We summarized the potential sources of methylation changes in a sentence,  “Thus, chemical, physical, nutritional, urbanistic and life style factors could affect DNA methylation” (Lines 143-146)

Page 5, line 210: “see (Schuettengruber et al., 2017)” Figure 1: The current linear depiction of maternal and paternal genomes in this Figure seems to imply the same genomes are inherited through generations, when really these genomes are from different individuals. The arrows at the top also seem overly simplistic for transgenerational exposures and inheritance. An improvement to this figure may be to divide the generations in to separate panels that can more accurately reflect transgenerational exposures and methylation, as well as different genomes.

We agree with the Reviewer’s comment. We have now clarified that different genomes are inherited through generations by adding a schematic representation of the mating on the right part of the figure. We also removed the arrows at the top of the figures (Please refer to modified Figure 1)

“Multigenerational exposures” across most of the top of the figure also conflicts with “environmental toxicant exposure” only indicated to affect the fetus or testis gonad.

We thank the reviewer for this comment. We meant “multigenerational effects”. This has been removed from the modified Figure 1.

Page 8, line 327: “Although epigenetic patterns are stable in somatic cells during adult life” is an inaccurate statement. Epigenetic patterns may be relatively stable compared to development, but there are epigenetic changes that can occur throughout life, which the authors previously describe through environmental exposures.

We agree with Referee. Indeed, we meant that global reprograming events occur in germ cells. In contrast, in differentiated cells, such as spermatocytes or spermatid fraction, the epigenetic changes were found at certain regions, but cells still retain the signature of germ cell lineage.  The statement has now been modified.

Page 8, lines 333-334: “In males, a third event occurs during spermatogenesis.” What are the first and second events?

The first event is the global epigenetic reprogramming that occurs during early embryonic development (in the preimplantation embryo) and the second one occurs during germline development from the PGC specification to the mitotic/meiotic arrest. Though we have mentioned this in the preceding sentence, we have now rephrased the statement to reflect clearly the first and the second reprogramming events. Also, we have indicated the embryonic days that correspond to the reprogramming events (Lines 347-349).

Page 9, lines 354-363: There is not a clear message of this paragraph. Do the authors believe epigenetic marks are not passed through generations?

This resetting process and its fidelity assure that the new lineages are well established and functional. It was found that Germline Reprogramming Responsive (GRR) genes could trigger the program of gonocytes in embryos. The epigenetic state at GRR genes is established by coordinated action of transcription factors and epigenetic modifier enzymes. Notably, some GRRs are involved in meiosis and gametogenesis, such as Dazl, Sycp1-3, Rad51c  [9]. Thus, somatic-to-germline reprograming is a comprehensive process underlying germline development, and any perturbation of this process likely affects not only current but also future generations (Lines 367-374).

The language in the last two sentences should be clarified and supported. What does “most epigenetic marks” and “their faithful preservation during global epigenetic reprogramming remains uncertain” mean?

Please refer to the previous response.

Page 9, line 400: The phrase “It has been known for a long time that” should be quantified or deleted.

The sentence has been rephrased as follows: “It has been known for more than 40 years that histones are not completely displaced” (Line 420).

Figures 1 and 2: the generational labels within each figure seem to be in conflict with one another. Figure 2 does not convey a clear message without reading the legend, in which case only the text should be used rather than a figure.

We removed Figure 2 and modified Figure 1. We have shifted the Figure 2 legend to text (Lines 480-487).

There is no depiction of the gametes that define F3 as the first unexposed generation and the red dashed arrows do not clearly show reduced epigenetic effects.

We have now removed Figure 2 from the manuscript.

References

Siklenka, K.; Erkek, S.; Godmann, M.; Lambrot, R.; McGraw, S.; Lafleur, C.; Cohen, T.; Xia, J.; Suderman, M.; Hallett, M., et al., Disruption of histone methylation in developing sperm impairs offspring health transgenerationally. Science 2015, 350, aab2006. Greer, E.L.; Maures, T.J.; Ucar, D.; Hauswirth, A.G.; Mancini, E.; Lim, J.P.; Benayoun, B.A.; Shi, Y.; Brunet, A., Transgenerational epigenetic inheritance of longevity in caenorhabditis elegans. Nature 2011, 479, 365-371. Dumasia, K.; Kumar, A.; Deshpande, S.; Balasinor, N.H., Estrogen, through estrogen receptor 1, regulates histone modifications and chromatin remodeling during spermatogenesis in adult rats. Epigenetics 2017, 12, 953-963. Andersen, H.R.; Andersson, A.M.; Arnold, S.F.; Autrup, H.; Barfoed, M.; Beresford, N.A.; Bjerregaard, P.; Christiansen, L.B.; Gissel, B.; Hummel, R., et al., Comparison of short-term estrogenicity tests for identification of hormone-disrupting chemicals. Environmental health perspectives 1999, 107 Suppl 1, 89-108. Okada, H.; Tokunaga, T.; Liu, X.; Takayanagi, S.; Matsushima, A.; Shimohigashi, Y., Direct evidence revealing structural elements essential for the high binding ability of bisphenol a to human estrogen-related receptor-gamma. Environmental health perspectives 2008, 116, 32-38. Lemaire, G.; Mnif, W.; Mauvais, P.; Balaguer, P.; Rahmani, R., Activation of alpha- and beta-estrogen receptors by persistent pesticides in reporter cell lines. Life sciences 2006, 79, 1160-1169. Sado, T.; Sakaguchi, T., Species-specific differences in x chromosome inactivation in mammals. Reproduction 2013, 146, R131-139. Hao, C.; Gely-Pernot, A.; Kervarrec, C.; Boudjema, M.; Becker, E.; Khil, P.; Tevosian, S.; Jegou, B.; Smagulova, F., Exposure to the widely used herbicide atrazine results in deregulation of global tissue-specific rna transcription in the third generation and is associated with a global decrease of histone trimethylation in mice. Nucleic acids research 2016, 44, 9784-9802. Hill, P.W.S.; Leitch, H.G.; Requena, C.E.; Sun, Z.; Amouroux, R.; Roman-Trufero, M.; Borkowska, M.; Terragni, J.; Vaisvila, R.; Linnett, S., et al., Epigenetic reprogramming enables the transition from primordial germ cell to gonocyte. Nature 2018, 555, 392-396. Nogues, P.; Dos Santos, E.; Jammes, H.; Berveiller, P.; Arnould, L.; Vialard, F.; Dieudonne, M.N., Maternal obesity influences expression and DNA methylation of the adiponectin and leptin systems in human third-trimester placenta. Clinical epigenetics 2019, 11, 20. Weng, X.; Liu, F.; Zhang, H.; Kan, M.; Wang, T.; Dong, M.; Liu, Y., Genome-wide DNA methylation profiling in infants born to gestational diabetes mellitus. Diabetes research and clinical practice 2018, 142, 10-18. Voisin, S.; Eynon, N.; Yan, X.; Bishop, D.J., Exercise training and DNA methylation in humans. Acta physiologica 2015, 213, 39-59.